# Extensive translation of small Open Reading Frames revealed by Poly-Ribo-Seq

**Julie L Aspden, Ying Chen Eyre-Walker, Rose J Phillips, Unum Amin, Muhammad Ali S Mumtaz, Michele Brocard, Juan-Pablo Couso***

School of Life Sciences, University of Sussex, Brighton, United Kingdom

**Abstract** Thousands of small Open Reading Frames (smORFs) with the potential to encode small peptides of fewer than 100 amino acids exist in our genomes. However, the number of smORFs actually translated, and their molecular and functional roles are still unclear. In this study, we present a genome-wide assessment of smORF translation by ribosomal profiling of polysomal fractions in *Drosophila*. We detect two types of smORFs bound by multiple ribosomes and thus undergoing productive translation. The 'longer' smORFs of around 80 amino acids resemble canonical proteins in translational metrics and conservation, and display a propensity to contain transmembrane motifs. The 'dwarf' smORFs are in general shorter (around 20 amino-acid long), are mostly found in 5'-UTRs and non-coding RNAs, are less well conserved, and have no bioinformatic indicators of peptide function. Our findings indicate that thousands of smORFs are translated in metazoan genomes, reinforcing the idea that smORFs are an abundant and fundamental genome component.

## Introduction

Small open reading frames (smORFs) of fewer than 100 amino acids exist in eukaryotic genomes in hundreds of thousands, but their annotation has been hindered by their size: short sequences are unable to obtain the high conservation scores that are the accepted indicator of functionality (*Ladoukakis et al., 2011*). This handicap, coupled with the limited numbers of experimentally proven functional smORFs in eukaryotes (*Kastenmayer et al., 2006*; *Galindo et al., 2007*; *Kondo et al., 2007*; *Hanada et al., 2012*; *Magny et al., 2013*; *Pauli et al., 2014*; reviewed in *Andrews and Rothnagel, 2014*), has so far precluded the reliable annotation of smORFs as coding sequences. Targeted bioinformatic approaches predict that hundreds of smORFs could be translated and functional in bacteria (*Hemm et al., 2008*), yeast (*Basrai et al., 1997*; *Kastenmayer et al., 2006*), plants (*Hanada et al., 2012*) and metazoans including *Drosophila* (*Ladoukakis et al., 2011*), mouse (*Frith et al., 2006*; *Crappe et al., 2013*), and humans (*Slavoff et al., 2013*), but this is in contrast to their low rate of detection in biochemical (proteomic) (*Falth et al., 2006*; *Slavoff et al., 2013*) and functional (genetic) (*Kumar et al., 2002*) screens. Thus, genome-wide assessments of smORF translation in a number of species are needed to determine the actual number of translated smORFs in eukaryotic genomes.

The new technique of ribosome profiling has corroborated and expanded the proteomes of yeast (*Ingolia et al., 2009*; *Duncan and Mata, 2014*; *Smith et al., 2014*), mouse (*Ingolia et al., 2011*), zebrafish (*Chew et al., 2013*; *Bazzini et al., 2014*), and *Drosophila* (*Dunn et al., 2013*). Thousands of new translated sequences have been described in each case, whether novel exons of annotated genes, alternative initiation sites, or entire ORFs. However, the application of ribosome profiling outside canonical translated sequences can lead to differing conclusions (*Chew et al., 2013*; *Guttman et al., 2013*). The problem remains that a ribosomal footprint cannot always be equated with translation; non-productive binding of single ribosomes to mRNAs and scanning 40S ribosomal subunits can result in footprints, yet do not constitute translation. It has been suggested that some smORFs associate

*For correspondence:
j.p.couso@sussex.ac.uk

**Competing interests:** The authors declare that no competing interests exist.

**Reviewing editor**: Thomas R Gingeras, Cold Spring Harbor Laboratory, United States

**eLife digest** To produce a protein, a stretch of DNA must first be transcribed to produce a molecule of messenger RNA (mRNA). The genetic information copied from the DNA is then read three letters at a time, in groups called codons. Each codon either encodes a particular amino acid to be added into a protein or provides further instructions: 'start codons' mark the beginning of a protein; 'stop codons' mark its end. The DNA between these two points is called an open reading frame (or ORF)—however, not all ORFs produce proteins.

Most proteins are made of several hundred amino acids, but the genomes of animals contain thousands of ORFs that would generate much smaller proteins made of fewer than 100 amino acids, if they were translated. It is, however, unclear how many of these small ORFs are converted into mRNA molecules and functional proteins.

Ribosomes are large molecular machines that translate the code in mRNA molecules and join together the appropriate amino acids in the right order to make a protein. Ribosome profiling is a technique that identifies which mRNA molecules are translated into proteins by determining the sequences of all the mRNA molecules bound to ribosomes at a particular moment. The mRNA sequences can then be compared with the sequence of the whole genome to work out which ORFs they correspond to. Ribosome profiling has been used to detect translated small ORFs, but the method yields a relatively high false positive rate as some mRNAs can bind to ribosomes without being translated.

To better detect small protein-producing ORFs, Aspden et al. developed a technique based on ribosome profiling called Poly-Ribo-Seq. The method takes advantage of the fact that during active translation, clusters of multiple ribosomes, called polysomes, bind mRNAs. Poly-Ribo-Seq isolates these polysomes and determines the sequence bound by each of the ribosomes, thereby reducing the number of false positives.

Applying Poly-Ribo-Seq to cells from the fruit fly *Drosophila* allowed Aspden et al. to identify two types of short ORF. The first type codes for proteins that are around 80 amino acids long and are translated with the same efficiency as larger ORFs. The sequences of these ORFs are found in other species, match at least in part sequences of known functional ORFs, and the proteins produced are found in specific locations inside cells. These small proteins may contribute to membrane integrity or function. Together, these properties suggest that these mRNAs create functional small proteins.

The second pool consists of very small ORFs ('dwarf smORFs') that code for around 20 amino acids, which are not translated so often and do not show many similarities with other species.

As the findings of Aspden et al. suggest that a large fraction of *Drosophila* small ORFs are translated into proteins, the next challenge will be to determine the roles of these small proteins in cells.

with ribosomes in such a non-productive manner and do not undergo productive translation (**Wilson and Masel, 2011**). Moreover, smORF mRNAs are short and present a small target for ribosomal binding and generation of footprints, potentially making traditional ribosome profiling less suitable for the study of smORFs. To distinguish genuine translation events from background, a number of metrics, statistical treatments, and bioinfomatic analyses have been proposed (**Ingolia et al., 2009**; **Chew et al., 2013**; **Guttman et al., 2013**; **Bazzini et al., 2014**). We have taken an alternative approach by enhancing the biochemical foundation of ribosome profiling and developed 'Poly-Ribo-Seq', an improvement to ribosome profiling that should be of use for the study of smORFs and canonical, longer ORFs alike. Instead of profiling all ribosomal-bound mRNAs, we perform ribosome footprinting on polysomal fractions. In this way, mRNAs bound by multiple ribosomes and hence actively translated can be isolated and distinguished from mRNAs bound by sporadic, putatively non-productive single ribosomes or ribosomal subunits. Although this method may overlook very short smORFs that cannot fit multiple ribosomes, this loss should be offset by the increased stringency of discarding false positive footprints.

Application of Poly-Ribo-Seq to *Drosophila* S2 cells reveals extensive translation of thousands of smORFs of two putative types. 'Longer' smORFs encode around 80-aa-long peptides, which are translated in the same proportions (83% of transcribed coding sequences) and as efficiently as canonical long ORFs. We nearly double the number of these smORFs shown to be translated, and show that these smORF peptides further resemble canonical proteins in that they are conserved across species,

show specific subcellular localisations, and display a specific amino acid composition with the potential to form transmembrane alpha-helices. 'Dwarf' smORFs encode peptides in 5'-UTRs and non-coding RNAs, are not detected by gene prediction programs, and are around 20-aa-long. They are less conserved, and translated at lower proportions and efficiencies than canonical proteins.

We corroborate these findings by two independent methods and observe smORF peptide expression at or near mitochondria. Extrapolation of our results suggests that thousands of smORFs are translated in higher organisms and that smORFs could have diverse functions, including expression of peptides active in cell membranes.

## Results

We have chosen to assess the translation of smORFs in *Drosophila*, because of the well-annotated genome of this organism and the availability of an equally well-characterised standard cell line (S2 cells) (*Schneider, 1972*) providing abundant and reproducible material.

The annotation of the *Drosophila* genome contains double the proportion of predicted smORF-encoding genes than other metazoan genomes (some 829 smORF genes, or 4% of the total, *Table 1*) (FlyBase, Ensembl). However, closer scrutiny reveals that although these genes have well-corroborated transcriptional data (modENCODE), less than a quarter of these have corroborated translation and peptide function. Only 164 annotated smORFs have at least two out of three markers indicating translation or peptide function: (1) molecular GO term indicating protein function (based on direct assays or presence of protein domains); (2) matches with peptides from proteomic experiments; and (3) conservation of the coding sequence beyond insects (*Figure 1A*). These 'corroborated' smORFs have in most cases a gene name (e.g., *Defensin*) and associated literature. The translation of the remaining 665 putative smORFs is thus not yet fully proven, and in most cases (494 smORFs) no evidence of translation is recorded. The majority of these 'uncorroborated' smORFs only have a cognate identifier (e.g., CG34200) and their 'coding' status varies between genome releases (unpublished observation). Thus, the *Drosophila* annotated smORFs offer an ideal framework to test for translation of smORFs and their biological importance.

### Development of 'Poly-Ribo-Seq', ribosome profiling of polysome fractions

Given the controversy over ribosome footprinting on lncRNAs (*Chew et al., 2013*; *Guttman et al., 2013*), we wanted to improve upon the ribosome profiling method to ensure that the RNAs, on which ribosome footprinting occurs, are undergoing active translation. That is to say they are engaged by polysomes rather than just bound by sporadic, putatively non-productive single ribosomes or ribosomal subunits. We therefore developed an approach for performing ribosome profiling on polysome complexes, using a modified ribosome footprinting method. Polysomal fractionation was used to separate RNAs, depending on the number and type of ribosomes bound to them (*Figure 1B*). In this way, mRNAs bound by multiple ribosomes and hence actively translated can be isolated and distinguished from mRNAs bound by non-productive 80S ribosomes. This biochemically purified material was then subjected to ribosome profiling, in which the footprinting reaction was optimized for profiling purified polysomal fractions rather than all ribosome-mRNA complexes.

To specifically enrich for actively translating single smORF-containing mRNAs, over canonical protein-coding mRNAs, we took advantage of the limited space within a smORF ($\leq$303 nt) for ribosomes to be bound. Ribosomes have been reported to reach densities of 1 ribosome every 80 nt (*Arava et al., 2003*), therefore on a smORF the maximum number of ribosomes associated would be five ribosomes (one at the start codon and one every 80 nt). Although RT-PCR of polysome sucrose

**Table 1.** Annotated smORFs in different organisms

|            | smORFs | ORFs    | % smORFs |
|------------|--------|---------|----------|
| Drosophila | 829    | 21,870  | 3.8      |
| Zebrafish  | 854    | 43,148  | 2.0      |
| Mouse      | 1131   | 51,745  | 2.2      |
| Human      | 1938   | 104,109 | 1.9      |

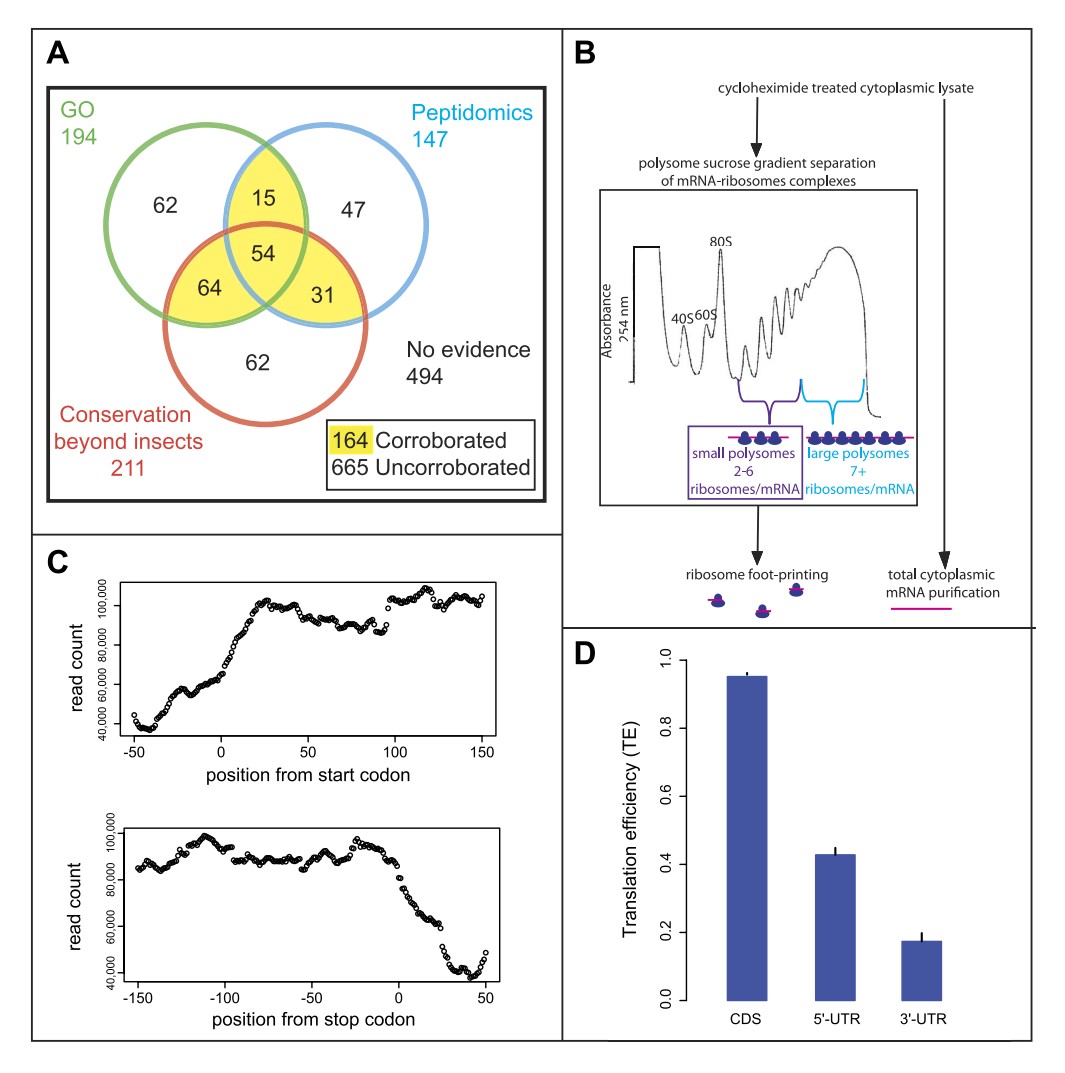

**Figure 1**. Poly-Ribo-Seq of small and large polysomes. (**A**) Venn diagram categorising annotated *Drosophila* smORFs as corroborated or uncorroborated based on evidence (FlyBase) from two out of three of: GO molecular function term assignment (green), peptidomic evidence (blue), and conservation outside of insects (red). Based on this, out of the total of 829 annotated smORFs, 665 are uncorroborated, and 494 have no evidence of translation. (**B**) Schematic of Poly-Ribo-Seq with representative UV absorbance profile for sucrose density gradient. Small (purple) and large (blue) polysomes are separated and subject to ribosome footprinting. (**C**) Composite plot from all FlyBase protein-coding genes of Poly-Ribo-Seq read counts across mRNAs in the vicinity of start (upper) and stop codons (lower) in small polysomes. (**D**) Median translational efficiencies of CDS, 5' and 3'-UTR regions for all protein-coding genes, error bars represent SE.

The following figure supplements are available for figure 1:

**Figure supplement 1**. Poly-Ribo-Seq of small and large polysomes.

**Figure supplement 2**. Schematic interpretation of Poly-Ribo-Seq.

gradient fractions confirmed that smORF mRNAs were enriched in small polysomes compared to large polysomes (*Figure 1—figure supplement 1A*), it suggested that *Drosophila* smORFs can be bound by up to six ribosomes, perhaps reflecting a tighter packing of ribosomes. We therefore chose to isolate polysomal fractions containing 2–6 ribosomes/mRNA to enrich for smORFs. These small polysomes can also contain mRNAs for canonical longer ORFs, which would be not fully covered by ribosomes and hence translated at less than maximum level (*Figure 1—figure supplement 2*).

We subjected both small and large polysomal fractions to ribosome profiling separately and performed RNAseq on the total cytoplasmic mRNA as a control (*Figure 1—figure supplement 1B*). Our 'Poly-Ribo-Seq' captured regions of active translation as ~80% of reads mapped to coding sequences of canonical protein-coding genes (*Supplementary file 1A*) with read densities dropping off before the start and after stop codons (*Figure 1C*). To quantify the translation of individual coding sequences we considered two metrics: (1) the ribosomal density in the ORF (expressed as RPKM) (*Ingolia et al., 2009*) and (2) coverage of the ORF by ribosome footprints (0–1). This metric indicates whether ribosomes bind across the ORF or just in a small fragment of it, which could be due to overlapping or internal ORFs (*Figure 1—figure supplement 2*). To be considered translated, we required ribosome densities to be at least 11.8 RPKM and footprint coverage of the ORF to be at least 0.57, which are above the 90th percentile of the values we obtained for the 3′-UTRs from canonical coding mRNAs (see 'Materials and methods' for a full explanation of filters and cut-offs). These cut-offs are more stringent than previous ribosomal profiling experiments and standard RNAseq practice, and their combination should provide robust identification of transcripts that undergo active translation. To overcome the possible dependence of ribosome density on RNAseq efficiency or transcript abundance (*Guttman et al., 2013*), we also used the relative metric known as translational efficiency (TE), which is the RPKM of ribosome footprints/RPKM of total mRNA control reads (*Ingolia et al., 2009*). We observed that the median TE of all annotated protein-coding transcripts was significantly higher in CDSs compared to 5′- and 3′-UTRs (*Figure 1D*) indicating that 'Poly-Ribo-Seq' defines regions of active translation. As previously reported for ribosomal profiling, we observe triplet phasing in the mapping of our Poly-Ribo-Seq reads (*Figure 1—figure supplement 1C*), reflecting the positioning of ribosomes on codons, which is not globally seen in UTRs (*Figure 1—figure supplement 1D*).

## Poly-Ribo-Seq detects smORF translation

Small and large polysomes showed a marked difference in genome-wide ribosomal densities (*Figure 2A*), suggesting that these two fractions contain mRNAs translated at different levels. Small polysomes contain mRNAs encoding long ORFs, but these have lower TE than when isolated from large polysomes (*Table 2*), confirming that they were bound by fewer ribosomes. As intended, Poly-Ribo-Seq detected smORFs with translation signatures, and these were enriched in small polysomes, which contained double the number and all of the smORFs detected in large polysomes (*Figure 2C*). The low smORF TE values in large polysomes are similarly consistent with low levels of smORF mRNAs being present in large polysomal complexes. However, the TE of smORFs from small polysomes is similar to the TE of long ORFs from large polysomes, indicating that smORFs can be translated at similar levels to standard protein-coding ORFs (*Table 2*; *Figure 1—figure supplement 2*).

Altogether 191 annotated smORFs passed the cut-off values to be deemed translated in this initial Poly-Ribo-Seq experiment. This is ~70% of the smORFs transcribed in S2 cells in the total mRNA controls (*Figure 2D*, small polysomes). To ensure that initial Poly-Ribo-Seq experiments sequenced to an adequate depth and to potentially extend the catalogue of translated smORFs, we repeated the experiment but exclusively sequenced small polysomes. This extensive small polysome profiling yielded nearly four times the number ORF-mapping reads obtained in the previous small polysome profiling (*Supplementary file 1A*) and detected translation of 224 smORFs (*Figure 2D*) expanding the number from both experiments to 227, which is 83% of smORFs we observed transcribed in S2 cells (*Figure 2E*). The genome-wide distribution of ribosome densities in two independent Poly-Ribo-Seq experiments was strongly correlated ($R^2$ = 0.83), suggesting that Poly-Ribo-Seq is highly reproducible (*Figure 2B*).

The majority of reads in both our experiments and previous ribosomal profiling consist of rRNA sequences released during footprinting (*Supplementary file 1A*; *Ingolia et al., 2011*). This reduces the depth of profiling and could preclude the detection of further smORFs. Therefore, we designed rRNA-depletion beads for use during footprint extraction ('Materials and methods', *Supplementary file 2*), which produced a marked improvement in the ratio of reads mapping to mRNAs (*Supplementary file 1A*) and increased the total number of ORFs detected (*Figure 2—figure supplement 1A*, −rRNA) but only expanded our overall catalogue of putatively translated smORFs by one (*Figure 2D*). The results of the three independent experiments are highly overlapping as 80% of putative translated smORFs were detected in all three data sets (*Figure 2D*). By combining the three experiments, we provide evidence that 228 smORFs are translated out of 274 transcribed in S2 cells (83%), which is very similar to the proportion of standard length protein-coding ORFs translated (81%) (*Figure 2E*). These

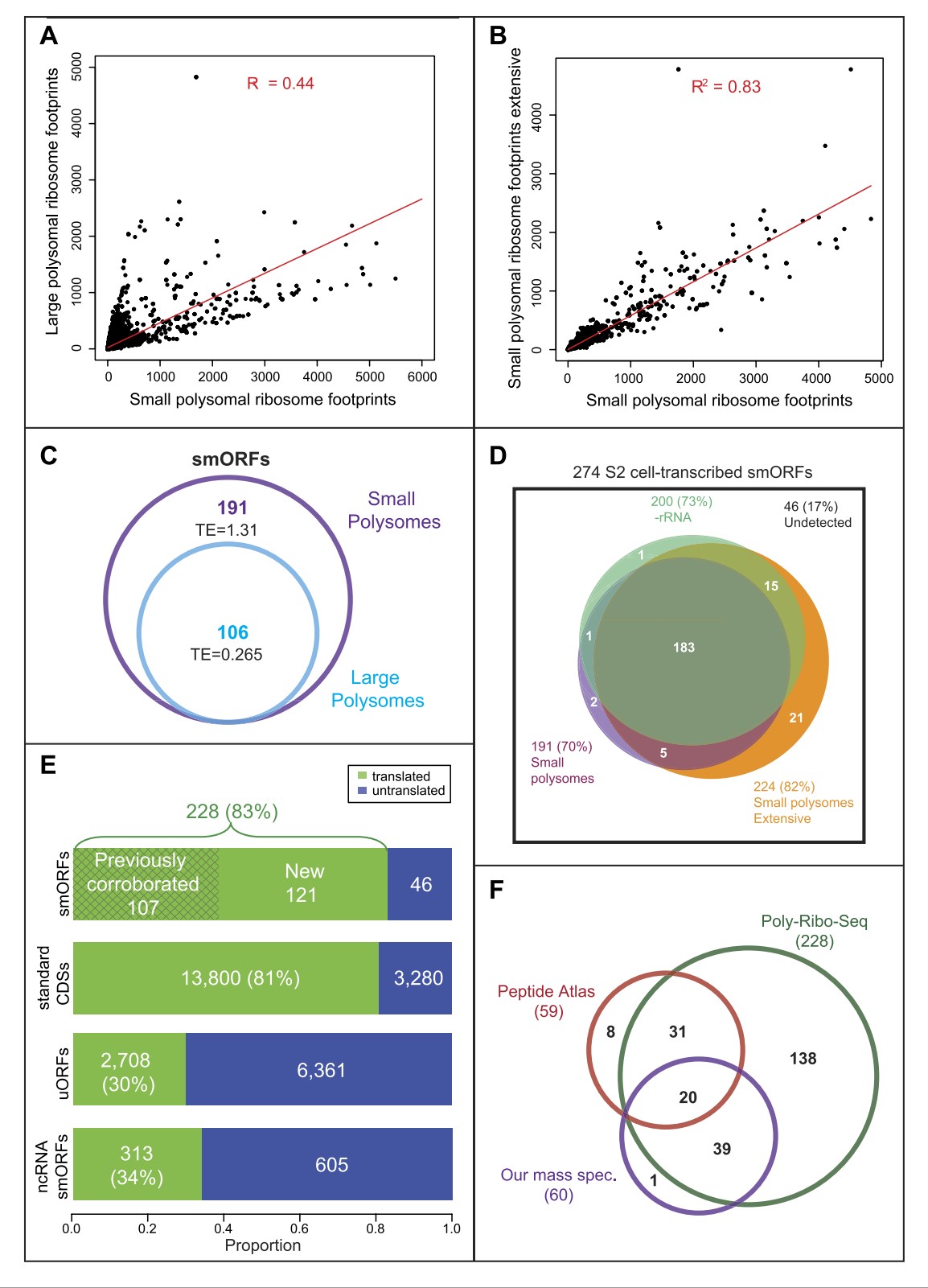

**Figure 2**. Poly-Ribo-Seq reveals translation of smORFs. (**A**) Ribosome footprinting densities (RPKM) from small polysomes correlate poorly with large polysomes (whereas two replicates of total cytoplasmic mRNA controls do, see **Figure 1—figure supplement 1B**). (**B**) Ribosome footprinting densities (RPKM) from small polysomes correlate highly between two biological replicates ($R^2$ = 0.83). (**C**) All 106 smORFs detected in large polysomes (blue) were

*Figure 2. Continued on next page*

*Figure 2. Continued*

also present in the 191 detected in small polysomes (purple). smORF footprints are much more abundant in small polysomes, as indicated by a higher TE value. (**D**) High coincidence of annotated smORFs detected as translated in three different Poly-Ribo-Seq experiments. Small polysome extensive experiment probes most deeply with 224 smORFs detected as translated (small polysomes: purple, small polysomes extensive: yellow, -rRNA: turquoise). (**E**) Numbers and proportions of transcribed ORFs, which are translated, according to Poly-Ribo-Seq data (translated: green, untranslated: blue). The proportion of annotated smORFs translated is similar to that of standard CDSs. 121 annotated smORFs are newly detected as translated, plus 2708 uORFs and 313 smORFs from ncRNAs. (**F**) Venn diagram showing overlap between Poly-Ribo-Seq (dark green), our mass spectrometry experiments (purple) and Peptide Atlas proteomic data (red).

The following figure supplement is available for figure 2:

**Figure supplement 1**. Poly-Ribo-Seq reveals translation of smORFs.

**Table 2.** Summary of median TEs

| Median TE | Small polysomes | Large polysomes |
|---|---|---|
| Annotated smORFs | 1.131 | 0.265 |
| standard ORFs | 0.829 | 1.110 |
| 5′-UTR | 0.355 | 0.566 |
| 3′-UTR | 0.162 | 0.196 |
| uORFs | 0.276 | 0.347 |
| ncRNA smORFs | 0.384 | 0.000 |

Median translational efficiency for ORFs in small and large polysomal fractions.

similar proportions may indicate the extent of translational regulation in S2 cells. Altogether this data almost doubles the previous repertoire of translated smORFs in *Drosophila* from 164 to 285 (164 previously corroborated [*Figure 1A*] and 121 new translated smORFs).

## Validation of smORF translation

The high overlap of our experiments suggests that the results do not arise from artefactual random sampling of smORFs, but most likely from the detection of the bona-fide population of annotated smORFs translated in S2 cells. To confirm this and independently validate our data, we compared our results with peptidomics data (Peptide Atlas, *Brunner et al., 2007*). Poly-Ribo-Seq increases nearly fourfold the number of smORFs with evidence of translation in S2 cells from 59 (Peptide Atlas) to 228 (Poly-Ribo-Seq). Poly-Ribo-Seq detects 86% of smORFs with Peptide Atlas evidence in S2 cells (51 out of 59 smORFs; *Figure 2F*), whilst only 8 smORFs, which have Peptide Atlas evidence are not shown to be translated by Poly-Ribo-Seq.

Detection of small peptides requires specific peptidomic methods (*Boerjan et al., 2010*; *Slavoff et al., 2013*), and this could have limited the number of smORF peptides detected in the generic proteomic experiments of Peptide Atlas. Therefore, we specifically searched for smORF peptides by performing mass spectrometry on two biological replicates of S2 cells after gel purifying small proteins 5 to 15 KDa in size, which corresponds to peptides predicted to be 45 to 130 aa in length. We detected a total of 60 annotated smORF peptides, of which 40 are not detected in Peptide Atlas S2 data sets (*Figure 2F*), thus bringing the combined pool of smORFs peptides detected by proteomics in S2 cells to 99 (*Figure 2—figure supplement 1B*). Despite this increase, Poly-Ribo-Seq was still more extensive. Poly-Ribo-Seq revealed 228 smORFs as translated, including 90 of the 99 in the combined proteomics pool (*Figure 2—figure supplement 1B*), and 59 of the 60 peptides detected by us (*Figure 2F*). The Poly-Ribo-Seq RPKM values of smORFs detected by peptidomics are over three times as high as those that are not (*Figure 2—figure supplement 1B*), suggesting that mass spectrometry detects peptides arising from the most highly translated smORFs, as also observed by other authors (*Brunner et al., 2007*; *Bazzini et al., 2014*).

To further validate the results of our smORF Poly-Ribo-Seq, we designed a peptide-tagging transfection assay. smORF coding sequences were tagged with a C-terminal FLAG tag lacking its own start codon. The constructs contained the full smORF 5′-UTR (which includes the Kozak and other sequences regulating translation [*Kozak, 2005*]) (*Figure 3—figure supplement 1A*). The resulting construct was transfected into S2 cells, where any FLAG signal would therefore be the result of smORF translation (*Figure 3—figure supplement 1A,B*). Transfection and staining of S2 cells with these FLAG-tagged smORFs confirmed the translation of all 12 smORFs tested, which exhibit a range of translational indicators (*Figure 3A,B*; *Table 3*) and peptidomics evidence, indicating that even lower levels of translation can give rise to smORF peptides detectable by this tagging method. Immunoblotting confirmed the

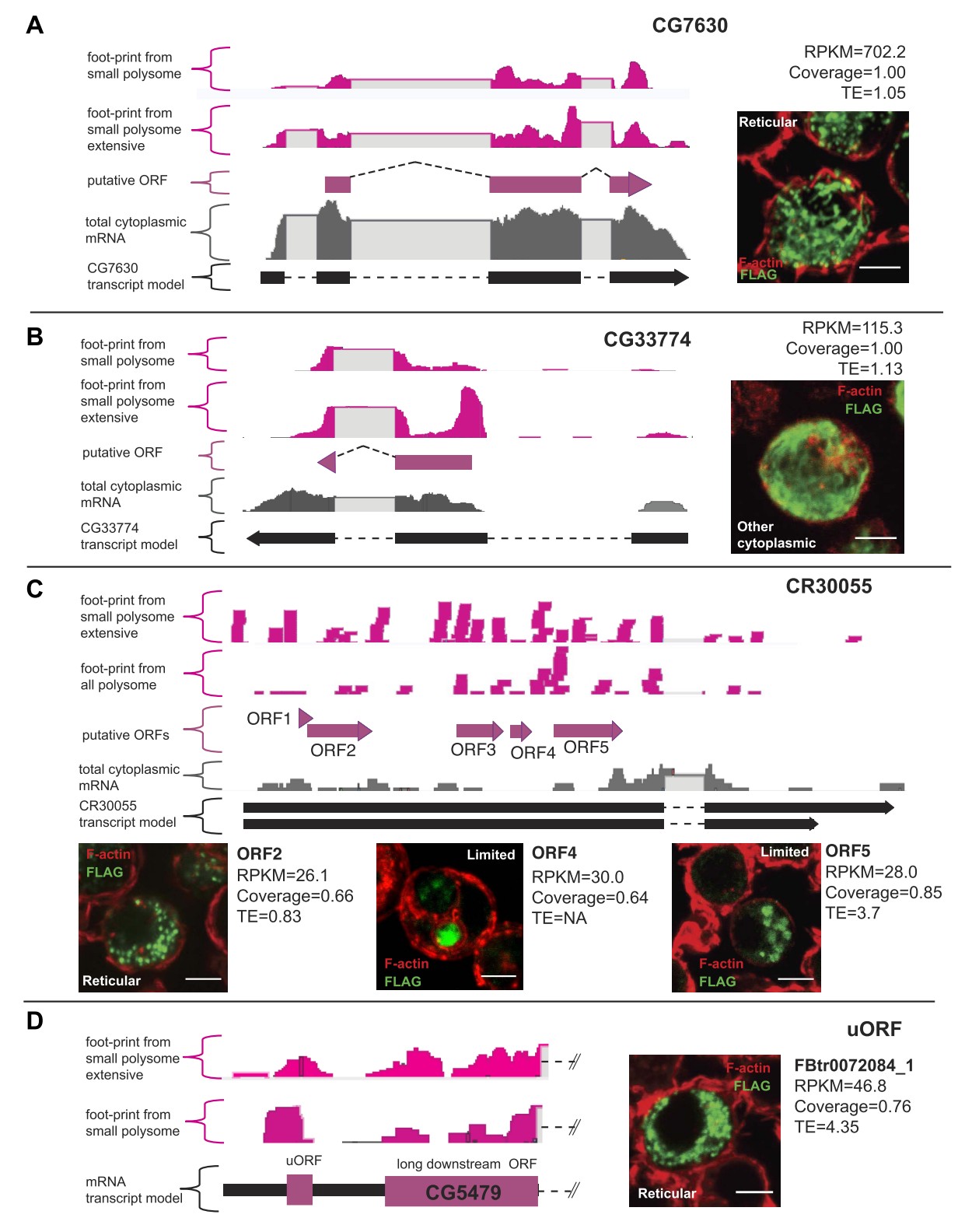

**Figure 3**. Validation of smORF translation by tagging assay. (**A–D**) Ribosome footprints from small polysomes (pink) and mRNA reads (grey) mapped to smORFs, along with transcript and ORF models of (**A**) CG7630, (**B**) CG33774, (**C**) CR30055 (ncRNA), and (**D**) FBtr0072084_1 (uORF). Corresponding transfection assays in S2 cells are shown (FLAG antibody: green, F-actin stained with phalloidin: red,

*Figure 3. Continued on next page*

*Figure 3. Continued*

scale bars = 5 µm) together with Poly-Ribo-Seq metrics (RPKM, coverage and TE). Distribution of each peptide (reticular, other cytoplasmic or limited) is indicated.

The following figure supplements are available for figure 3:

**Figure supplement 1**. Validation of smORF translation by tagging assay.

**Figure supplement 2**. Poly-Ribo-Seq reveals translation of ORFs in ncRNAs.

**Table 3.** Summary of tagged annotated smORFs

|  | Localization | Peptidomic evidence | # aa | RPKM | Coverage | TE | Phast Cons |
|---|---|---|---|---|---|---|---|
| **CG32230** | **Mitochondrial** | **Yes** | **83** | **539.2** | **1.00** | **3.05** | **0.54** |
| **CG14482** | **Mitochondrial** | **Yes** | **57** | **600.0** | **1.00** | **1.09** | **0.72** |
| CG44242 | Mitochondrial | Yes | 70 | 152.9 | 0.97 | 1.75 | 0.66 |
| CG7630 | Mitochondrial | Yes | 90 | 702.2 | 1.00 | 1.05 | 0.64 |
| CG33199 | Mitochondrial | No | 79 | 95.5 | 1.00 | 1.17 | 0.59 |
| CG32582 | Mitochondrial | No | 52 | 16.5 | 0.57 | 2.82 | 0.51 |
| **sclA** | **Other cytoplasmic** | **NA** | **28** | **NA** | **NA** | **NA** | **NA** |
| **CG12384** | **Other cytoplasmic** | **Yes** | **96** | **205.6** | **1.00** | **1.37** | **0.71** |
| CG33774 | Other cytoplasmic | No | 40 | 115.3 | 1.00 | 1.13 | 0.73 |
| CG33170 | Other cytoplasmic | No | 71 | 84.2 | 0.84 | 0.75 | 0.60 |
| CG34200 | Limited | Yes | 52 | 331.7 | 1.00 | 1.66 | 0.54 |
| CG32267 | Limited | Yes | 49 | 82.5 | 0.97 | 1.13 | 0.70 |
| CG33155 | Limited | No | 60 | 33.8 | 0.64 | 0.88 | 0.67 |
| tal-B | None | NA | 49 | NA | NA | NA | NA |

Details of the Poly-Ribo-Seq and transfection translation assay results for the FLAG-tagged smORFs, with RPKM, coverage and TE values. Previously corroborated smORFs (according to **Figure 1A**) are in bold. Scl is a positive control and tal-B is a negative control, but both are not endogenously transcribed in S2 cells, hence 'NA' Polysomal Ribo-Seq metrics and Peptidomic evidence.

expected sizes of the tagged peptides (*Figure 3—figure supplement 1C*). Tagged peptides exhibit distinct subcellular localisations, which are suggestive of different peptide functions (*Figure 3A,B*; *Table 3*). Six smORFs display a reticular distribution resembling mitochondria, an inference supported by their co-localization with the mitochondrial marker Mitotracker Red (*Figure 3A* and *Figure 3—figure supplement 1D,E*; *Table 3*) and the available information from homologues of two 'corroborated' smORFs in this group, CG32230 and CG14482 (*Tripoli et al., 2005*). Six smORFs exhibited other types of anisotropic cytoplasmic localisation, similarly to ER-expressed Sarcolamban smORF (*Magny et al., 2013*), indicating that they may localise to other cytoplasmic compartments (*Figure 3B* and *Figure 3—figure supplement 1D,E*; *Table 3*).

The putative functionality of smORF peptides is further supported by over half of the translated smORFs having revealed a function in high-throughput S2 cell RNAi screens in previous studies (*Schmidt et al., 2012*; *Figure 3—figure supplement 1F*). The biological relevance of smORFs is also implied by the transcription of 196 of these translated smORFs in embryos (*Supplementary file 1B*). 88 of these smORFs are transcribed throughout the whole of embryogenesis, which might be indicative of a basic cellular or housekeeping role, whereas 47 have stage-specific expression, perhaps indicative of a developmental role.

## Other sources of smORFs

Non-annotated smORFs were also scrutinized by Poly-Ribo-Seq. Many putative ncRNAs have been annotated as such because no long ORFs have been detected, but they can still contain smORFs. We

looked for evidence of translation in 6438 ORFs that initiate with an AUG start codon within ncRNAs. Our total cytoplasmic mRNA data indicate that 125 ncRNA transcripts (containing 918 different ORFs) are transcribed and present in the cytoplasm of S2 cells. 313 smORFs in these transcripts appear translated by Poly-Ribo-Seq (*Figure 2E*), but ncRNA smORFs behaved differently from protein-coding and smORF genes. The median translation efficiency of these putative smORFs within ncRNA genes is lower than for canonical genes and annotated smORFs, and in fact is similar to UTRs (*Table 2*). In addition, we could not observe nor obtain peptidomics corroboration for the encoded peptides, and the FLAG signal is limited for the majority of such smORFs tested in the transfection assay (*Figure 3C*, *Figure 3—figure supplement 1G*). Yet a sizeable fraction (34%) of non-coding RNA smORFs displayed Poly-Ribo-Seq metrics above our cut-off values indicating translation (*Figure 2E*; *Table 4*) and some can display FLAG and Western blot signal similar to annotated smORFs (*Figure 3C*, *Figure 3—figure supplement 1G,H*; *Table 4*). Further, these positive-testing smORFs from non-coding RNAs show codon read-phasing (*Figure 3—figure supplement 2A*). These translation events do not necessarily represent 'background' translation of non-coding RNAs associated with polysomes. The comparison of ribosomal footprinting reads with reads resulting from the sequencing of RNA from polysomal fractions before footprinting (as in polysomal profiling) shows a high correlation for canonical coding sequences as expected (*Smith et al., 2014*), but not for non-coding RNAs, where high RNASeq polysomal counts do not necessarily result in significant footprinting (*Figure 3—figure supplement 2B*). Altogether, our results suggest that a proportion of these so-called non-coding RNA genes actually contain smORFs that are actively translated in S2 cells.

Upstream short ORFs, or uORFs, have been described in more than 50% of annotated mammalian transcripts encoding canonical, long ORFs (*Fritsch et al., 2012*). We identified 14,881 uORFs with AUG start codons, within 11,587 5'-UTRs of 28,529 FlyBase annotated transcripts. 9069 of these uORFs were transcribed in S2 cells and of these 2708 (30%) are footprinted by ribosomes (*Figure 2E*). Similarly to smORFs in putative non-coding RNAs, translated uORFs display lower median TE than canonical ORFs and translated smORFs (*Table 2*), and they are not detected by peptidomics, altogether suggesting low abundance of the encoded peptides. However, tagging of uORFs can occasionally show similar signal to annotated smORFs (*Figure 3D*, *Figure 3—figure supplement 1I*).

## Bioinformatic analysis of translated smORFs reveals specific characteristics

We scrutinised our set of annotated translated smORFs for bioinformatic markers, which might further suggest function of smORF peptides. Firstly, we used phastCons (*Siepel et al., 2005*) that measures conservation between 12 insect species.

**Table 4.** Summary of tagged smORFs from non-coding RNAs and uORFs

| smORF | Localization | Peptidomic evidence | # aa | RPKM | Coverage | TE | PhastCons |
|---|---|---|---|---|---|---|---|
| pncr009:3L ORF1 | Other cytoplasmic | No | 21 | 135.7 | 1.00 | 0.29 | 0.44 |
| pncr009:3L ORF2 | Limited | No | 30 | 64.7 | 0.58 | 0.63 | 0.49 |
| pncr009:3L ORF3 | Limited | No | 33 | 47.8 | 0.78 | 0.23 | 0.59 |
| CR30055 ORF1 | Not tested | No | 12 | 15.2 | 0.71 | 1.24 | 0.49 |
| CR30055 ORF2 | Mitochondrial | No | 53 | 26.1 | 0.66 | 0.83 | 0.52 |
| CR30055 ORF3 | Not tested | No | 36 | 54.6 | 0.85 | 2.90 | 0.55 |
| CR30055 ORF4 | Limited | No | 17 | 30.0 | 0.64 | NA | 0.54 |
| CR30055 ORF5 | Limited | No | 56 | 28.0 | 0.85 | 3.7 | 0.55 |
| Uhg2-ORF 1 | None | No | 36 | 10.5 | 0.27 | 0.83 | 0.54 |
| FBtr 0072084_1 | Reticular | No | 14 | 46.8 | 0.76 | 4.35 | 0.52 |
| FBtr 0072210_1 | Other cytoplasmic | No | 13 | 97.7 | 0.92 | 4.34 | 0.48 |
| FBtr 0081720_1 | Limited | No | 11 | 121.3 | 1.00 | 2.39 | 0.55 |

Details of the Poly-Ribo-Seq and transfection translation assay results for the FLAG-tagged smORFs translated from non-coding RNAs and uORFs, with RPKM, coverage and TE values.

We examined the phastCons values in intergenic sequences and canonical long protein-coding sequences (*Figure 4A*) and obtained a cut-off value of 0.55 separating them (10% FDR). 93% of S2-translated smORFs have phastCons scores above this threshold (median = 0.66), indicating a conservation level similar to that of canonical long-ORFs, and hence, a similar level of functionality for the coding sequences.

As a further indicator of smORF translation, we studied the amino acid composition of translated smORFs, compared to canonical long proteins and expected random usage (*Figure 4B*). Annotated smORFs display a lower than random usage of arginine, which is a hallmark of translated proteins (*King and Jukes, 1969*). However, they also display differential usage of several amino acids, which are characteristic of alpha-helices in canonical proteins, being enriched for lysine and phenylalanine, and depleted of serine (*Chou and Fasman, 1974*; *Figure 4B*, *Figure 4—figure supplement 1A*). This finding was corroborated by an abundance of putative transmembrane alpha-helix motifs, in about a third of translated smORFs (*Figure 4C*) and all predicted smORFs (*Figure 4D*) compared to the expected 20% observed in canonical proteins (*Krogh et al., 2001*). This is in agreement with similar findings in bacteria (*Hemm et al., 2008*) and suggests that smORFs may represent a source of uncharacterised transmembrane peptides. An enrichment for molecular GO terms such as membrane transporter activity in annotated smORFs (*Figure 4—figure supplement 1B*), and the subcellular localisations we observe for half of the tagged smORFs, are also consistent with these findings.

The 555 annotated smORFs not transcribed in S2 cells, including 505 smORFs with uncorroborated translation, share the bioinformatic characteristics of the smORFs detected as translated by Poly-Ribo-Seq including: average peptide length (*Figure 4E–F*); amino acid usage (*Figure 4B*, *Figure 4—figure supplement 1A*); and abundance of putative transmembrane alpha-helices (*Figure 4C,D*). Therefore, our results showing translation and possible peptide function for 83% of the smORFs transcribed in S2 cells could be extrapolated to this wider pool, potentially bringing the number of smORFs encoding functional peptides in *Drosophila* to around 700.

The peptides encoded by uORFs and ncRNAs did not behave bioinformatically as smORFs and canonical long CDS, and thus we cannot easily extrapolate from our results to the uORFs and ncRNAs not transcribed in S2 cells. No indicator (phastCons, size, aa composition) was able to distinguish translated uORFs and ncRNAs from intergenic or random sequences (*Figure 4—figure supplement 1C–D*). Even though our Poly-Ribo-Seq detects translation of these uORFs and ncRNAs, perhaps the function of these smORFs is not mediated by their encoded peptides, or at least is compatible with a shorter, more variable and less canonical amino acid sequence ('Discussion'). It appears that Poly-Ribo-Seq detects translation of two types of smORF: (a) 'longer' smORFs that are efficiently translated producing peptides ~80 aa possessing bioinformatic hallmarks of peptide function; and (b) 'dwarf' smORFs translated from ncRNAs and 5'-UTRs, which are in general shorter (~20 aa), less efficiently translated, and missing such bioinformatic and molecular markers.

## Discussion

### smORFs are translated in high numbers in metazoans

We have developed an improvement to ribosome profiling, which we term Poly-Ribo-Seq, to ensure that footprinted mRNA sequences represent regions of active translation rather than non-productive events. Using Poly-Ribo-Seq, we have specifically profiled the translation of smORFs in *Drosophila* S2 cells, using a purification of small polysomes to enrich for smORFs.

Poly-Ribo-Seq doubles the number of annotated smORFs in *Drosophila* S2 cells with evidence of translation from 107 to 228. Translated smORFs seem similar to canonical proteins, both in terms of the fraction actually translated (over 80% in both cases, *Figure 2E*) and the level of translation (as revealed by RPKM, coverage, and TE). Extrapolating the proportion of smORFs translated in S2 cells (83%) to the uncorroborated 544 smORFs transcribed elsewhere, indicates that altogether around 700 annotated smORFs could be translated, which can be tested in further Poly-Ribo-Seq experiments.

The annotation of the *Drosophila* genome is unusual in its high proportion of annotated smORFs, which is double that of vertebrate genomes (*Table 1*). Thus, examination of smORF translation in vertebrate genomes is likely to significantly increase the proteome of these species. Consistent with this prediction the translation of 190 smORFs has been detected in one such experiment in zebrafish (*Bazzini et al., 2014*). Extrapolation of these two sets of results in flies and zebrafish strongly suggests that hundreds of smORFs are translated in higher organisms, including in humans and other mammals.

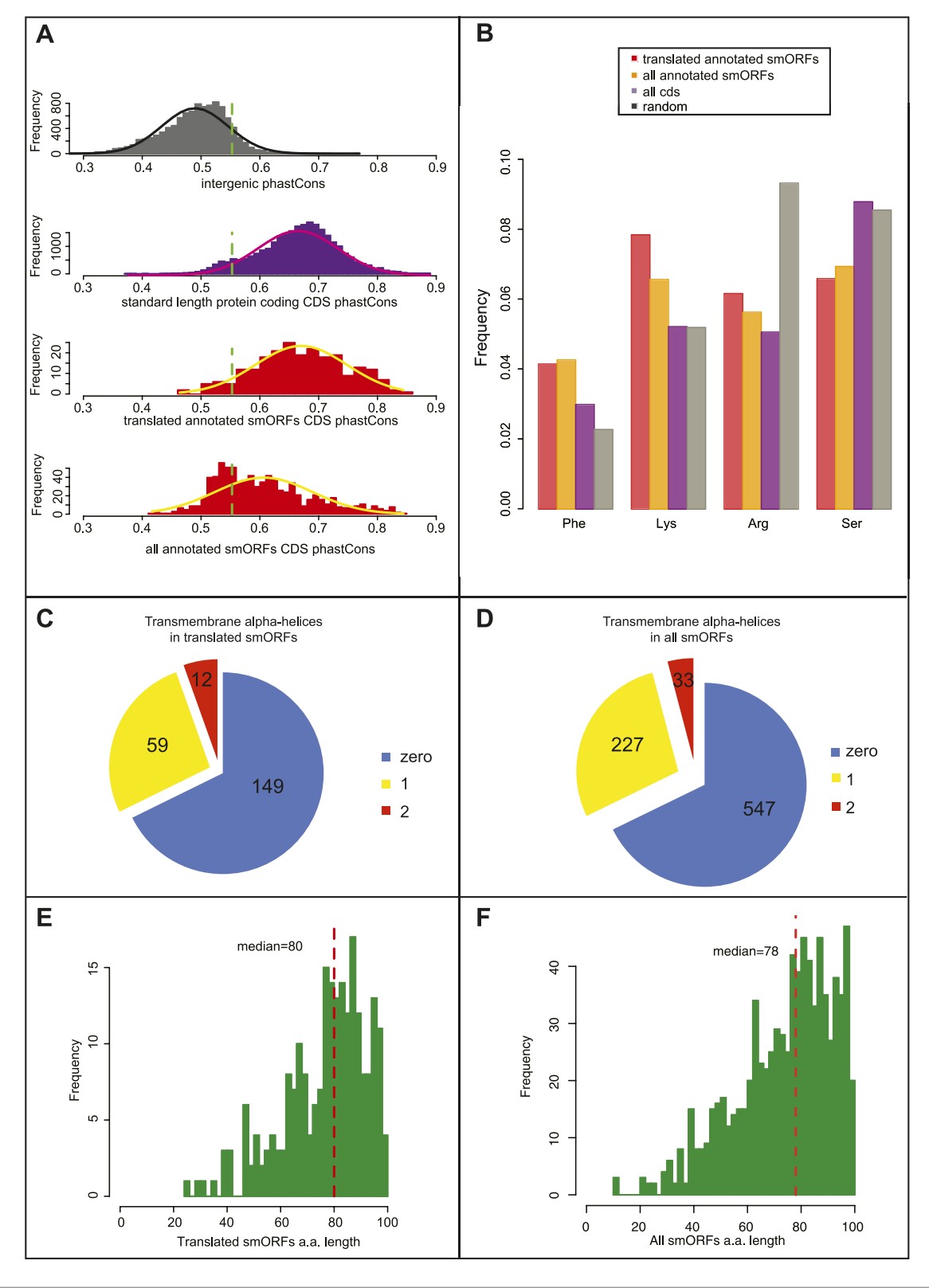

**Figure 4**. Bioinformatic indicators of smORFs. (**A**) Distribution of phastCons scores for intergenic regions, standard length protein-coding CDSs (longer than 100 aa), S2 cell-translated annotated smORFs, and all annotated smORFs, with fitted normal curves. Green dotted lines indicate the 90th percentile of intergenic phastCons scores (0.55). (**B**) Relative abundance of particular amino acids in proteins (random expected: black, all CDSs: purple, all annotated

*Figure 4. Continued on next page*

*Figure 4. Continued*

smORFs: yellow, and translated smORFs: red). (**C** and **D**) Proportion of (**C**) S2-cell translated (32%) and (**D**) all smORFs (32%) predicted to contain transmembrane α helices (TMHMM). (**E** and **F**) Frequency distribution of smORF peptide lengths for (**E**) translated and (**F**) all annotated smORFs with medians shown by red dotted line.

The following figure supplement is available for figure 4:

**Figure supplement 1**. Bioinformatic indicators of smORFs.

While bioinformatics can be useful in unearthing smORF candidates, ultimately experimental evidence and full functional characterisation is the only way to ascertain the translation and function of each individual smORF.

Our data confirm the tentative annotation of *Drosophila* smORFs and expand our understanding of smORF translation. Firstly, we corroborate the translation of 121 smORFs for which no previous conclusive evidence existed. Secondly, we fail to detect the translation of some 46 transcribed smORFs, most of which have no evidence of translation, and which (pending new profiling experiments from further biological sources), could be either translationally regulated or non-functional. Thirdly, we also detect the translation of a high number of non-annotated smORFs from 5′-UTRs and non-coding RNAs.

313 ncRNA smORFs and 2708 uORFs are detected as translated by Poly-Ribo-Seq, although perhaps at low levels, as indicated by their low median TEs. Given the small proportion of non-coding RNAs we detect in S2 cells, it is unclear if the numbers observed here can be extrapolated to the smORFs found in all other ncRNAs. However, we notice that if extrapolated to other animals, such as mammals including humans, even a low number of translated ncRNAs would be significant. The human genome contains some 32,000 transcripts currently annotated as long non-coding RNAs (*Volders et al., 2013*), and a fraction around 44% of long-non-coding RNAs have been detected in the cytoplasm (*Derrien et al., 2012*). If these cytoplasmic lncRNAs were corroborated and translated in the same proportion as we find here in *Drosophila* (some 34%), there could be thousands of human peptides awaiting detection and characterisation. The number swells to many thousands when adding uORFs. The small size of both uORF and ncORF peptides may hinder their detection by mass spectrometry, as in the case of peptides translated by the *Drosophila* genes *tal* and *scl* (unpublished observation), and so absence of proteomic detection (or absence of functional data, see below) should not be used to disprove their translation.

Poly-Ribo-Seq detects almost all peptides detected by proteomics, but is two to three times more extensive. Furthermore, Poly-Ribo-Seq can define the whole of the translated ORF as opposed to isolated micropeptides detected by peptidomics. Though available proteomic evidence has been useful in confirming the depth of Poly-Ribo-Seq, it is clear that currently peptidomics is not as sensitive as ribosome profiling. High peptide translation level seems a critical factor (but likely not the only one) favouring detection by mass spectrometry. However, the combination of Poly-Ribo-Seq and peptidomics could produce interesting data on peptide stability and degradation.

## Function of smORFs: translation and beyond?

The putative function of smORFs and their encoded peptides is a separate issue from their translation, just as the transcription of thousands of apparently non-coding RNAs is an accepted fact separated from their, as yet, not fully understood function. Our present work is concerned with proving smORF translation, as a first step to eventually uncovering their true function. However, the function of a number of smORFs has been identified in animal and plants (reviewed in *Andrews and Rothnagel, 2014*), and our data allow for some speculations.

We observe that most annotated and translated smORFs have conservation levels similar to canonical proteins and have so far displayed functionality in RNAi tests in some 50% of the cases. They encode peptides of around 80 aa with a high proportion of transmembrane alpha-helix motifs, in agreement with their overall pattern of amino acid composition. Their bioinformatic indicators (conservation, size, aa usage, and transmembrane motifs) appear similar in the 505 uncorroborated smORFs not transcribed in S2 cells, suggesting that, if translated, some 700 smORFs might encode peptides with similar functional potential. The corroborated smORFs display a variety of functions as antibacterial peptides (*Lemaitre and Hoffmann, 2007*), cell signals (*Pueyo and Couso, 2008*), cytoskeletal regulators (*Djakovic et al., 2006*), and other regulators of canonical proteins (*Hanyu-Nakamura et al.,*

*2008*; *Magny et al., 2013*). GO term enrichment analysis shows that the most abundant terms amongst previously corroborated smORFs are ribosome constituents, oxidoreductase activity, and transporter activity (*Figure 4—figure supplement 1B*). The last two imply an association with biological membranes, and it is thus interesting that amongst uncorroborated smORFs, the frequency of predicted transmembrane alpha-helix motifs is the highest. Furthermore, the observed tagged smORF peptide localization to mitochondria is also compatible with these findings. We surmise that hundreds of small, membrane-associated peptides are awaiting characterisation, and they may alter our understanding of many cellular and organismal processes of biological and medical relevance. We expect that these peptides would interact with canonical proteins as their regulators (*Magny et al., 2013*), as their size limits their structural capabilities; though, such capabilities could be expanded by oligomerization.

The main described function of most uORFs is to regulate translation of downstream long ORFs, either interfering with the translation of the downstream ORF, or collecting ribosomes in order to promote it. The translated uORF peptide (and hence its aa sequence) can be irrelevant ([*Child et al., 1999*; *Kulkarni et al., 2011*] reviewed in *Kozak, 2005*; *Andrews and Rothnagel, 2014*). Such a *cis*-regulatory role would fit with the low TE and the high sequence variability (low phastCons scores) observed for uORFs. However, we do not observe an overall positive or negative correlation between the translation of uORFs and that of their main downstream ORF (unpublished observation), as perhaps could be expected under this *cis*-regulatory model. Validating uORF function is difficult, since much of the current functional (genetics) data is based on conditions (gene deficiencies, promotor mutations, RNAi) that knock-out whole transcripts, that is, both the short uORF peptide and the downstream long-canonical protein.

The smORFs in putative long non-coding RNAs with ribosomal signatures might reveal either coding potential, a non-coding association with ribosomes, or a dual function as coding and non-coding for these transcripts. Again, standard genetic techniques are unable to distinguish between these possibilities. This group of smORFs has the potential to encode functional peptides, as shown by the genes *tal* and *scl*, that were previously annotated as putative non-coding RNAs. These genes encode peptides as short as 11 and 28 aa with important functions in development and physiology (*Galindo et al., 2007*; *Kondo et al., 2007*; *Magny et al., 2013*). In the case of Scl, a transmembrane alpha-helical structure was corroborated (*Magny et al., 2013*). Further, a manual study of their homologies identified their conservation in distant species. Thus, a case-by-case bioinformatic and experimental examination of ncRNA smORFs may reveal unknown numbers of new bioactive peptides.

Altogether our data indicate that thousands of smORFs are translated in metazoan genomes. However, they also suggest the existence of two broad types of translated smORFs. The 'longer' smORFs produce conserved 80 aa-long peptides whose translation efficiencies resemble those of canonical proteins and with functions biased towards an association with cell membranes. The 'dwarf' smORFs are mostly found in 5'-UTRs and non-coding RNAs, are not detected by gene prediction programs, and on average encode peptides of some 20 aa-long. They are also less conserved, translated at lower efficiencies, and of unclear function as yet.

## Materials and methods

### Tissue culture

S2 cells were grown under standard conditions in Schneiders medium with 10% FBS.

### Poly-Ribo-Seq

S2 cells were treated with cycloheximide (Sigma, St Louis, MO) at 100 µg/ml for 3 min at RT before harvesting. The cells were pelleted, washed (1X PBS, 100 µg/ml cycloheximide), and resuspended in lysis buffer; 50 mM Tris–HCl pH8, 150 mM NaCl, 10 mM MgCl$_2$, 1 mM DTT, 1% NP40, 100 µg/ml cycloheximide, Turbo DNase (Life Technologies, Carlsbad, CA), RNasin Plus RNase Inhibitor (Promega, Carlsbad, CA), cOmplete Protease Inhibitor (Roche). Nuclei were removed, and cytoplasmic lysates were loaded onto sucrose gradients and subjected to ultracentrifugation. Gradients were pumped out, their absorbance at 254 nm plotted and fractionated. We purified mRNAs in small polysomes, away from monosomes (80S), ribosomal subunits (40S, 60S), and large polysomes. Footprinting was performed overnight at 4°C with RNaseI (Life Technologies), stopped with SUPERase·In RNase inhibitor (Life Technologies) and precipitated. mRNA from total cytoplasmic lysate was purified using oligo (dT) Dynabeads (Life Technologies) and fragmented by alkaline hydrolysis. 28–34 nt ribosome

footprints and 50–80 nt mRNA fragments were gel purified and prepared as previously described (*Ingolia et al., 2009*, *2011*, *2012*) for Next Generation Sequencing. Libraries were sequenced on Illumina HiSeq2000 and MiSeq machines with 50 bp SingleEnd read protocol.

## rRNA depletion

To generate ssDNA complementary to *Drosophila* rRNA, PCRs were performed using 5′ biotinylated reverse primers (*Supplementary file 2*). A 5′ biotinlyated oligo complementary to 2S rRNA and rRNA PCR products were bound to magnetic streptavidin beads (Life Technologies) and their second strands washed away. Two rounds of 50 µl rRNA beads were used to deplete rRNA prior to reverse transcription.

## RT-PCR

RNA from sucrose gradient fractions was precipitated with isopropanol and 0.3 M NaCl. Resuspended pellets were treated with Turbo DNaseI (Life Technologies), extracted with phenol/chloroform and re-precipitated. cDNA was synthesised MMLV reverse transcriptase (Promega) and subjected to PCR with mRNA specific primers and Taq Polymerase (Qiagen, Venio, Netherlands).

## Footprint sequence alignment

Sequencing reads were clipped, trimmed, and aligned to an rRNA and tRNA reference using Bowtie, discarding the rRNA and tRNA alignments and collecting unaligned reads. Unaligned reads were mapped to FlyBase (Release 5.50) using TopHat. We only retained reads that mapped uniquely, but allowed up to two mismatches.

## Footprint profile analysis

Profiles of ribosome footprints across a transcript were constructed by quantifying the number of footprint reads aligned at each position within the feature of interest. Ribosome density was computed by scaling read counts for each feature-by-feature length and by the total number of genome-aligned reads (*Ingolia et al., 2009*). Footprint coverage estimated the percentage of each feature covered by ribosome footprints using the BEDTools coverageBed command.

We applied several filters to ascertain translation. We have taken the reads in the 3′-UTRs of mRNAs encoding canonical coding sequences (longer than 100 aa), from the small polysomal fraction, as representing 'background', that is likely non-coding sequences from mRNAs that are lowly translated. We obtained the RPKM and coverage values from this canonical 3′-UTR signal, and use their 90th percentile values as preliminary cut-offs for accepting translation of coding sequences. These values are 11.8 RPKM and 0.40 coverage. Superimposed onto these we use two additional corrections: first, we raised the coverage cut-off to 0.57 because 'dwarf' smORFs of less than 20 aa (*Figure 4—figure supplement 1C*) containing a single ribosomal binding 'site' could still appear as 0.56 covered (32 nt/57 nt); second, we introduce the need for an ORF to obtain at least five reads in a single experiment to be considered translated, to avoid inflation of very few reads by the RPKM metric in such dwarf smORFs.

Translation efficiency (TE) was calculated as ribosome footprint density (RPKM)/mRNA-seq read density (RPKM) in the feature. As the TE score is not a reliable estimator at low expression levels, we computed a TE score only for those features that had significant mRNA expression above a randomised genomic background ($p < 0.01$).

To analyse framing, ribosome-protected fragments (RPF) were aligned to transcript cooordinates. For a given open reading frame, the corresponding P-site position of filtered RPF reads (28–32 nt) was designated as follows: +12 offset for 28 and 29 nt, +13 for 30 to 31 nt, and +14 for 32 nt RPF (*Chew et al., 2013*; *Bazzini et al., 2014*).

## uORF and ncORFs identification

We identified uORFs and ncORFs longer than 10 aa with an AUG start codon followed by an in-frame stop codon within the annotated 5′-UTRs and ncRNA transcripts, using the emboss getorf program. To exclude the possibility that the ribosome occupancy observed in 5′-UTRs was due to the presence of such upstream ORFs, we created a modified transcript that contained all regions except the putative uORFs for all our analysis on 5′-UTRs.

## phastCons values

phastCons scores for 171,317 alignment blocks were downloaded from UCSC Genome Browser. We computed percentage overlap between the phastCons block and our feature of interest and estimated mean phastCons values.

## Peptide Atlas

Lists of peptide CDS coordinates with protein identifiers (FlyBase peptide ID) were downloaded from Peptide Atlas database (http://www.peptideatlas.org) and compared to FlyBase predicted smORF peptide sequences.

## Functional analysis of smORFs

Prediction of transmembrane alpha-helices was performed using TMHMM (http://www.cbs.dtu.dk/services/TMHMM/). In house perl scripts (available in *Supplementary file 3*) calculated amino acid composition of CDS. For the random control, we followed *King and Jukes (1969)*. We took all FlyBase transcript sequences and calculated the nucleotide composition of this pool; from this we estimated the likely amino acid usage based on the nucleotide composition of the respective codons. RNAi screen data were accessed through Flymine (http://www.flymine.org/), and GO term enrichment was calculated by GOrilla (http://cbl-gorilla.cs.technion.ac.il/) (*Eden et al., 2009*).

## Cloning

The 5′-UTR and CDS of putative smORFs were cloned by PCR from S2 cell cDNA into pENTR/D-TOPO (Invitrogen) and then into pAWF (http://emb.carnegiescience.edu/labs/murphy/Gateway%20vectors.html#), whose ATG start codon was mutated to GCG by site-directed mutagenesis.

## Transfections and microscopy

S2 cells were plated on acid-treated coverslips and transfected with plasmid DNA using Xtreme Gene HP (Roche). After 48 hr, the cells were fixed for 20 min with 4% formaldehyde, washed with 1X PBS, 0.1% Triton X–100 (PBS-T), blocked with PBS-T 2% wt/vol BSA before immunostaining with primary mouse anti-FLAG M2 antibody (Sigma) at 1/1000 and secondary anti-mouse FITC (Jackson, West Grove, PA) at 1/400. For subcellular localisation experiments, cells were incubated for 30 min with Rhodamine-Phalloidin (Life Technologies) to highlight F-actin. All transfections were incubated for 10 min with Hoechst (Sigma) according to manufacturer's instructions for nuclei staining and mounted with Vectashield (Vector Labs, Burlingame, CA). For Mitotracker experiments, 48 hr after transfection cells were incubated in 500 nM Mitotracker Red CMXRos (Life Technologies) for 45 min. Imaging was conducted using a Zeiss 63X Plan Apochromat Oil Immersion lens on the LSM510 Axioskop 2. For correlation analysis Z-stack images were taken with slice interval of 0.15 µm, and ImageJ plugin 'Manders Coefficients' was used to calculate correlation coefficient of FLAG to Mitotracker signal with at least 15 cells per replicate, with three replicate transfections.

## Immunoblotting

Cells were harvested 48 hr post transfection, washed with 1X PBS, resuspended in Tricine Sample buffer (Bio-Rad, Hercules, CA) (2.5% vol/vol βME) and run on 16% Tris-Tricine gels. Immunoblots were incubated with primary antibody: 1:10,000 anti-FLAG M2 (Sigma) and 1:500 anti-β-tubulin E7 (DSHB, Iowa City, IA), and then secondary 1:10,000 goat anti-mouse HRP (Santa Cruz, Dallas, TX). Immunoblots were developed with ECL Prime Chemiluminescent Detection Reagent (GE Healthcare, Little Chalfont, UK).

## Mass spectrometry

S2 cells were lysed in 0.075% SDS, 1X c0mplete Protease Inhibitor Cocktail (Roche) with three rounds of freeze thawing and clarified. Total protein (1X Tricine Loading buffer, 2.5% vol/vol βME) was run on 10–20% MiniProtean Tris-Tricine Gels (Bio-Rad) and the 5–15 KDa region excised. Mass spectrometry was performed by Cambridge Centre for Proteomics (University of Cambridge, UK) using in-gel trypsin digestion and LC-ESI-MS/MS using an Orbitrap Velos Instrument (Thermo Fisher Scientific) with the following parameters: 2 missed Trypsin cleavages, 25 ppm Precursor mass error, 0.8 Da fragment mass tolerance, carbamidomethylation of cysteine as a fixed and methionine oxidation as a variable modification. Spectra were matched against *Drosophila melanogaster* (5.55) proteome using generic Mascot algorithm.

## Acknowledgements

We thank Simon Morley and Nick Ingolia for help with protocols, Claudio Alonso, Jose Ignacio Pueyo, and Emile Magny for manuscript comments. This work was funded by a Wellcome Trust Senior Fellowship (ref 087516).

# Additional information

### Funding

| Funder | Grant reference number | Author |
|---|---|---|
| Wellcome Trust | 087516 | Juan-Pablo Couso |

The funder had no role in study design, data collection and interpretation, or the decision to submit the work for publication.

### Author contributions

JLA, Conception and design, Acquisition of data, Analysis and interpretation of data, Drafting or revising the article; YCE-W, Acquisition of data, Analysis and interpretation of data, Drafting or revising the article, Contributed unpublished essential data or reagents; RJP, Acquisition of data, Contributed unpublished essential data or reagents; UA, Acquisition of data, Analysis and interpretation of data, Drafting or revising the article; MASM, Acquisition of data, Analysis and interpretation of data; MB, Conception and design, Contributed unpublished essential data or reagents; J-PC, Conception and design, Analysis and interpretation of data, Drafting or revising the article

# Additional files

### Supplementary files

• Supplementary file 1. (**A**) Summary of sequencing experiments. Number of reads; from each experiment, that are left after removal of rRNA and tRNA contaminants, that are unique matches and that map to CDS regions of the genome. (**B**) Summary of smORF embryo RNA-seq data. Number of translated smORFs expressed throughout embryonic stages of *Drosophila melanogaster,* according to RNAseq data (modENCODE).

• Supplementary file 2. Primers used for rRNA depletion.

• Supplementary file 3. In house Perl scripts.

### Major datasets

The following dataset was generated:

| Author(s) | Year | Dataset title | Dataset ID and/or URL | Database, license, and accessibility information |
|---|---|---|---|---|
| Aspden JL, Eyre-Walker YC, Couso JP | 2014 | Extensive translation of small Open Reading Frames revealed by Poly-Ribo-Seq | http://www.ncbi.nlm.nih.gov/geo/query/acc.cgi?acc=GSE60384 | Publicly available at NCBI Gene Expression Omnibus. |

The following previously published dataset was used:

| Author(s) | Year | Dataset title | Dataset ID and/or URL | Database, license, and accessibility information |
|---|---|---|---|---|
| The Flybase Consortium | 1999 | Flybase | http://flybase.org/ | http://flybase.org/wiki/FlyBase:About#FlyBase_Copyright. |

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
