## [Decision Letter]

Thank you for sending your work entitled “Extensive translation of small ORFs revealed by Polysomal Ribo-Seq” for consideration at *eLife.* Your article has been evaluated by Chris Ponting (Senior editor), a Reviewing editor, and 3 reviewers, two of whom are members of our Board of Reviewing Editors.

The Reviewing editor and the other reviewers discussed their comments before we reached this decision, and the Reviewing editor has assembled the following comments to help you prepare a revised submission.

This manuscript by Aspden et al. entitled “Extensive translation of small ORFs (smORF) revealed by polysomal Ribo-seq” is an exploration into the use of only portions of long non-coding (lnc-) or coding RNAs as template for translation. The reviewers agreed with both the timeliness and importance of this topic and its potential significance. The manuscript is written in a relatively clear manner but in several portions is sparse in needed information and clarity (see below). While there is much to recommend this manuscript, several issues will need to be addressed to assist the reader to better understand or accept the findings.

Major issues:

1) The authors emphasize how this new method (Polysomal Ribo-Seq) represents a significant improvement from the previous approach. They should compare their approach to the recently published ribosome footprinting in *Drosophila* method by the Weissman laboratory to indeed determine whether it makes any difference to isolate polysomal fractions and thus test their claim.

2) The authors seem to choose cutoffs quite arbitrarily. There are four general areas that might be addressed and related to this topic:

a) It is unclear whether they have tested the performance of these cutoffs on known coding genes. “To be considered translated, we required ribosome densities to be above 7.7 RPKM and footprint coverage of the ORF to be above 0.57”. The authors should benchmark their methods against known translated sequences.

b) What is needed is the comparison of RNAseq data to peptide (mass spec) sequencing. Values of 7.7 RPKM and 57% coverage should be compared to translation of known protein coding RNAs (especially small proteins, e.g., Dm HSP22 (177 aa), Dm HSP 23, DM HSP 26, etc).

c) For Figure 2, it could be that the R^2 value of small vs large may largely be driven by large polysomal mRNAs with very low RPKMs compared to the smORF RNAs. What levels of expression (RPKM) were used as a threshold to be placed into this plot? Is it not possible that many of the large polysomal RNAs are not different but are merely expressed at a lower level vs the small polysomal level?

d) Validation of ribosomal profiling data suffers from the issue of incompleteness. Detection of Atlas peptides is not fully explained. What percent of peptides predicted by RNAseq matches the atlas peptide; how many atlas peptides/smORF are detected?

3) The interpretation of the results for the ncRNA smORFs is not the most parsimonious. The entire argument for their translatability rests on the association of the lncRNas with the polysomal riboseq. This raises the issue of whether the detection of RNAs in this fraction is truly indicative of translatability. In the absence of the other criteria mentioned by the authors (mass spec and conservation) the co-fractionation of the RNA with polysomes can be also attributable to the non-specific affinity of the ribosome proteins to RNA. For example, can the authors show that a metagene of the lncRNAs translated show phasing of the ribosome in the ORFs undergoing translation?

4) Regarding uORFs, the authors state that 3,404 (38%) uORF are footprinted by ribosomes (Figure 2). This seems a surprisingly high number, specially taking into account that they find fewer than 300 sORF translated. Are the authors using the same criteria to detect translated uorfs for small ORFs? Is one footprint enough? What criteria are used when there are overlapping ORFs in different reading frames? How do the authors distinguish between two overlapping ORFs to determine which one is indeed being translated?

5) The data should be made accessible, including sequences/genome coordinates of ORFs identified as well as upstream ORFs. In the current manuscript this data is not accessible except for a few examples (Table 3).

6) While the selection of RNAs bound to 2-6 ribosomes is reasonable, what is the relationship of the lengths of the RNAs found to be bound compared to the coverage by the ribosomes? Are these RNAs also short or are they present in long (i.e., ∼1.5-2 Kb) RNAs of which only ∼300 nt are covered? Were annotated mRNAs found in this category that could be been translated in a different reading frame for a short length?

---

## [Author Response]

*1) The authors emphasize how this new method (Polysomal Ribo-Seq) represents a significant improvement from the previous approach. They should compare their approach to the recently published ribosome footprinting in Drosophila method by the Weissman laboratory to indeed determine whether it makes any difference to isolate polysomal fractions and thus test their claim*.

We argue that our improvement is not a quantitative, but a qualitative one, and this has been re-emphasized in the Introduction. Because our ribosomal profiling reads come from an mRNA with several ribosomes attached to it, one can be more certain that such reads and ribosomal binding indicate productive translation. If anything, our method could be expected to be more astringent than standard ribosomal profiling as it should exclude (or dramatically reduce) reads due to non-productive 80S and 40S binding.

Regarding Weissman’s lab data (14), it is difficult to compare experiments using different techniques but we have compared our data with their S2 cell data. They do not present RNASeq controls for their S2 experiment, therefore certain comparisons are not possible. Nonetheless, if we apply our filters and cut-offs to their data, we observe that more smORFs would appear as translated in their data (264 theirs vs. 228 ours). Therefore our method is either a) less extensive, or b) more astringent than standard ribosomal profiling (for the detection of smORFs). We favour b) since these 36 Weissman-specific smORFs have low RPKMs and only 2 are detected by proteomics. We surmise that these Weisman-specific smORFs may represent either smORFs lowly expressed in our S2 cells, or non-productive ribosomal binding (40S and 80S binding). In fact 14 of them don’t pass our transcription cut-off of 1 RPKM in S2 cells. This is not a criticism of their work, but in our opinion (and others) an unavoidable feature of the standard ribosomal profiling method, and the very reason and justification for our modification.

2) The authors seem to choose cutoffs quite arbitrarily. There are four general areas that might be addressed and related to this topic:

*a) It is unclear whether they have tested the performance of these cutoffs on known coding genes. “To be considered translated, we required ribosome densities to be above 7.7 RPKM and footprint coverage of the ORF to be above 0.57”. The authors should benchmark their methods against known translated sequences*.

*b) What is needed is the comparison of RNAseq data to peptide (mass spec) sequencing. Values of 7.7 RPKM and 57% coverage should be compared to translation of known protein coding RNAs (especially small proteins, e.g., Dm HSP22 (177 aa), Dm HSP 23, DM HSP 26, etc)*.

We appreciate these suggestions, which we have evaluated, and that are similar to other ideas we considered in the past.

There are in general two ways of finding cut-off values:

A) Starting with positive controls, find a value that eliminates the lowest possible percentile of positive controls (in our case, could be 10% of translated ORFs) while eliminating a reciprocal, or very substantial, percentile of negative controls.

B) Starting with negative controls, find a value that eliminates the highest possible percentile of negative controls (in our case, 90% of non-coding mRNA sequences) while eliminating a reciprocal, or minimal, percentile of positive controls.

Regarding A): these heat-shock proteins suggested by the referees are lowly transcribed and translated in S2 cells under standard conditions and therefore their use as benchmarks for cut-offs would considerably reduce the astringency of our experiment. We have tried, as an alternative, small ribosomal proteins, but these are translated at very high levels and hence produce too astringent cut-offs, that would discard most canonical proteins. To avoid the subjectivity involved in selecting this or that set of small proteins as benchmarks one could use all smORFs to obtain cut-offs, but this would produce a circular argument (if no smORFs were really translated, their coding sequences would produce very low metrics but still would have a top 90th percentile).

Alternatively, we could use a percentile of the values from canonical long ORFs as benchmarks, but this would imply a pre-judgment of the fraction of these long ORFs that we believe to be subjected to translational regulation (that is mRNA is transcribed, but not translated); would this be 10% for a 90th percentile cut-off? Or 25% for a 75th percentile?). It also entails the assumption that smORFs are going to be translated at similar strength as canonical proteins, a premise that is in fact under test, and therefore can’t be accepted a priori.

To use proteomics-detected peptides as a benchmark for cut-offs does not work either. We have re-written the section in the manuscript that compares and validates our results with proteomics to make it clearer. Our results and those of others indicate that proteomics only detects smORFs with higher profiling metrics (in other words, more strongly translated). Hence, to use proteomics-corroborated smORFs to obtain cut-offs would preclude the detection of smORFs translated less strongly, and will negate the main point of doing ribosomal profiling to begin with: to obtain a more extensive, yet sufficiently astringent, repertoire of translated smORFs.

In synthesis, in this experiment one does not have independent positive controls: all positives are part of the sample and subject to test.

Therefore, we used the strategy B) above, the highest percentile of likely negative controls, in our case 3’ UTRs, in accordance with other ribosomal profiling papers. However, we take the point of using only known translated genes, as this avoids circular arguments as discussed above for smORF coding sequences. Using this strategy, note for example how the fraction of annotated (longer) smORFs deemed translated by these cut-offs is very similar to the fraction of translated canonical genes (Figure 2). Because the cut-offs were not selected from these smORFs, or even from canonical ORFs, but independently from canonical UTRs, this is a non-circular and non-trivial result.

Thus, we have revised our cut offs and re-extracted them using only canonical coding genes as suggested:

We have taken the reads in the 3’-UTRs of mRNAs encoding canonical coding sequences (longer than 100aa) from the low polysomal fraction in our first experiment. These mRNAs should be translated at a low level since they have fewer ribosomes attached to them, and in fact their TE and RPKMs are lower. Hence, reads in their 3’-UTRs represent our best source of independent negative controls (or ‘background’): non-coding sequences that should not show meaningful ribosomal binding, from mRNAs that are lowly translated to begin with. We obtain the RPKM and coverage values from these canonical 3’-UTR reads, and use their 90th percentile value as preliminary cut-offs for accepting translation of coding sequences. These values are 11.8 RPKM (which is considerable more astringent than the usual cut-off of RPKM > 1) and 0.40 coverage (a metric that is astringent and useful in its own way, see later on).

Superimposed to these 90th 3’-UTR percentile values, we use two additional corrections. To avoid the case where a very short ORF of less than 20aa could contain a single ‘site’ of ribosomal binding, and to better differentiate between overlapping ORFs, the 0.40 coverage was raised to 0.57. For the analysis of ‘dwarf’ smORF profiling (see discussion of major issue 4 below) we also used an additional filter. A minimum of 5 reads in a single experiment was demanded, to avoid again situations where a few reads on a very short ORF get artificially inflated to a high RPKM. A full explanation of these filters and cut-offs has been added to the Methods section.

The astringency of our new filters is shown by the borderline case of CG15456, which now falls just short of the new cut offs (with an RPKM of 10.1) but however we observed to produce weak but reproducible signal in transfection assays by immunofluorescence and immunoblotting, and would pass our cut-offs using the [14] data. Thus, we may have increased the likelihood of false negatives, but reciprocally, we should have reduced the likelihood of false positives. A truly negative result, the long-non-coding RNA Uhg2, has now been added to the results in Figure 3—figure supplement 1 and Table 4 for comparison.

*c) For*
Figure 2*, it could be that the R^2 value of small vs large may largely be driven by large polysomal mRNAs with very low RPKMs compared to the smORF RNAs*. *What levels of expression (RPKM) were used as a threshold to be placed into this plot? Is it not possible that many of the large polysomal RNAs are not different but are merely expressed at a lower level vs the small polysomal level?*

Yes the mRNAs in small and large polysomes are not different but are footprinted at different levels, this is what we intended to show. No threshold was used in this graph (perhaps the high overlap of low values makes it look like it was).

The same long canonical mRNAs produce more reads in large polysomes, expressed as higher RPKM (Figure 1—figure supplement 2 vs Figure 1—figure supplement 2) and TE (Table 2), than in small polysomes. This fits with the expectation that mRNAs have more ribosomes bound when they are being translated at a higher level.

*d) Validation of ribosomal profiling data suffers from the issue of incompleteness. Detection of Atlas peptides is not fully explained*. *What percent of peptides predicted by RNAseq matches the atlas peptide; how many atlas peptides/smORF are detected?*

These data were included in Figure 2—figure supplement 1. However, several queries focus on this proteomics section, so we have tried to clarify it by changing the text in the ‘Validation of smORF translation’ section, and by swapping Figure 2 with Figure 2—figure supplement 1, and eliminating Figure 2—figure supplement 1. To answer this specific question, 51 out of 59 Peptide Atlas peptides detected in S2 cells are also detected by our Poly-Ribo-Seq (86%). 51 peptides represent 22% of our translated smORFs. In our own proteomics experiments, we detect 60 peptides of which 59 are also detected by Poly-Ribo-Seq. Altogether, 99 peptides are detected by proteomics in S2 cells, and of these, Poly-Ribo-Seq detects 90 (91%); the total percentage of smORF thus corroborated by proteomics is 39%. The peptides detected by proteomics, as mentioned in minor issue 2 below, seem translated at higher levels, and hence perhaps are more abundant.

*3) The interpretation of the results for the ncRNA smORFs is not the most parsimonious. The entire argument for their translatability rests on the association of the lncRNas with the polysomal riboseq. This raises the issue of whether the detection of RNAs in this fraction is truly indicative of translatability. In the absence of the other criteria mentioned by the authors (mass spec and conservation) the co-fractionation of the RNA with polysomes can be also attributable to the non-specific affinity of the ribosome proteins to RNA. For example, can the authors show that a metagene of the lncRNAs translated show phasing of the ribosome in the ORFs undergoing translation*?

We now provide the framing analysis for the translated smORFs in the two non-coding RNAs highlighted in our figures (Figure 3—figure supplement 2). Note however, that for unknown reasons framing is not as good in Drosophila as in other organisms (see Figure 1—figure supplement 1 and Dunn, Weisman 2014). This is one of the reasons why we have introduced the coverage metric. Notice also that translation is also corroborated by tagging experiments in Figure 3 and Figure 3—figure supplement 1.

The possibility of ‘non-specific ribosomal protein binding’ is present, but is reduced by:

a) the discard of reads shorter than 25 nt.

b) the selection (via polysomes) against mRNAs just bound by single ribosomes, or partial ribosomes, or other proteins.

c) filter for 5 reads and 0.57 coverage.

We have now added another control that illustrates the specificity of Poly-Ribo-seq footprinting (Figure 3—figure supplement 2). We have sequenced RNAs that are associated with 2-6 polysomes before footprinting, and correlated it with the footprints we later observe. As expected, in general there is a good correlation for putative coding transcripts (annotated smORFs and canonical proteins) such that, if an RNA is present in a polysomal fraction, it is bound by ribosomes and translated. However, for non-coding RNAs, this correlation is much weaker and in fact, below our footprint RPKM cut-off for accepting translation, a number of lncRNAs can be present in the polysomal fraction in high amounts, yet do not give rise to ribosomal footprints. Hence, association with polysomes does not necessarily mean translation. Association with polysomes and significant generation of footprints, does.

*4) Regarding uORFs, the authors state that 3,404 (38%) uORF are footprinted by ribosomes (*Figure 2*). This seems a surprisingly high number, specially taking into account they find fewer than 300 smORF translated. Are the authors using the same criteria to detect translated uORFs for small ORFs? Is one footprint enough? What criteria are used when there are overlapping orfs in different reading frames? How do the authors distinguish between two overlapping ORFs to determine which one is indeed being translated*?

We had applied the same criteria for uORFs as for all other ORFs. However, we had raised the coverage cut-off to 0.57 because of these uORFs, based on the following calculation: the median size of apparently translated uORFs is 57 nucleotides (19 aa, see Figure 4—figure supplement 1). A bona-fide ribosomal binding site could be as long as 32 nucleotides; hence, a uORF containing a single ribosomal binding ‘site’ could still be 0.56 covered (32nt/57nt). But in addition, the referees identify here another important technical point. The RPKM metric inflates artificially the number of reads in very short sequences, and for example, an 11.8 RPKM cut-off could still leave some dwarf smORFs of around 20aa being considered translated with as little as 2 reads. We have therefore added a further filter for these ‘dwarf’ smORFs (uORFs and non-coding smORFs) of needing a minimum of 5 reads in a single experiment to be considered translated, in addition to the RPKM and coverage filters we apply to all cds. For reference, the lowest number of reads obtained with an annotated smORF is 8. Since these annotated smORFs are on average longer (hence able to generate more reads) and have higher translational efficiencies than dwarf smORFs, a cut-off filter of 5 reads for the latter (the midpoint between 2 and 8) seems reasonable.

The new and more astringent filters specifically impinge on dwarf smORFs, as shown by a reduction in their numbers from the previous version of the manuscript. Annotated smORFs and standard long cds deemed translated have dropped by 2-3% (from 86 to 83% and from 83 to 81% respectively), whereas uORFs and ncRNA-ORFs drop by 8-9% (from 43 to 34% and from 38 to 30% respectively).

Despite these corrections (higher RPKM cut-off, new filter of 5 minimum reads) and a corresponding reduction in the number of translated uORFs, 2,708 still pass our filters (some 30% of the total). We also find these numbers intriguing, but other authors have also found a very high number of uORFs apparently translated (Chew 2014, [4], [13]). We would like to stress here, as in the paper, that this translation does not equate to peptide function, and that the role of many uORFs could be simply cis-regulatory. However, it is possible that some of these uORFs produce stable and functional peptides. As a further test, we have tagged several uORFs and the results corroborate the profiling data (see new Figure panels Figure 3 and Figure 3—figure supplement 1).

Finally, our metric used to distinguish which overlapping smORF is translated is coverage. By requiring 0.57 coverage, we greatly increase the likelihood of distinguishing overlapping ORFs, although some overlapping and apparently translated ORFs exist (and can be independently discerned as translated by virtue of their non-overlapping reads). We have added a new figure (Figure 1—figure supplement 2) that clarifies these scenarios.

*5) The data should be made accessible*, *including sequences/genome coordinates of ORFs identified as well as upstream ORFs. In the current manuscript this data is not accessible except for a few examples (*Table 3*)*

We have uploaded fastq and other data files as suggested in the GEO website.

*6) While the selection of RNAs bound to 2-6 ribosomes is reasonable, what is the relationship of the lengths of the RNAs found to be bound compared to the coverage by the ribosomes? Are these RNAs also short or are they present in long (i.e., ∼1.5-2 Kb) RNAs of which only ∼300 nt are covered? Were annotated mRNAs found in this category that could be been translated in a different reading frame for a short length*?

In general there is no clear correlation between the length of the mRNAs and the length covered by ribosomes. We understand here that the referees may be worried by the possibility that reads are generated by a small fraction of a long ORF, thus possibly by a ‘hidden’ smORF within such long ORFs. This would be scenario D) in the new Figure 1—figure supplement 2. This scenario is discarded by the coverage metric. High coverage ensures that when mRNAs encoding canonical ORFs longer than 100aa and detected in the 2-6 ribosome fraction are called translated it is because, as discussed in point 2 above, the canonical ORF is being translated uniformly but at low level, rather than only a small portion or different frame of it.